# Evidence-Based Dietary Practices to Improve Osteoarthritis Symptoms: An Umbrella Review

**DOI:** 10.3390/nu15133050

**Published:** 2023-07-06

**Authors:** Ashley N. Buck, Heather K. Vincent, Connie B. Newman, John A. Batsis, Lauren M. Abbate, Katie F. Huffman, Jennifer Bodley, Natasha Vos, Leigh F. Callahan, Sarah P. Shultz

**Affiliations:** 1Kinesiology Department, Seattle University, Seattle, WA 98122, USA; anbuck@unc.edu; 2Department of Exercise and Sport Science, University of North Carolina, Chapel Hill, NC 27402, USA; 3Osteoarthritis Action Alliance, Chapel Hill, NC 27599, USA; hkvincent@ufl.edu (H.K.V.); cncbn@optoline.net (C.B.N.); john.batsis@unc.edu (J.A.B.); lauren.abbate@va.gov (L.M.A.); katie_huffman@med.unc.edu (K.F.H.); nvos@unca.edu (N.V.); leigh_callahan@med.unc.edu (L.F.C.); 4Department of Physical Medicine and Rehabilitation, College of Medicine, University of Florida, Gainesville, FL 33865, USA; 5Division of Endocrinology, Diabetes and Metabolism, Department of Medicine, New York University School of Medicine, New York, NY 10016, USA; 6Division of Geriatric Medicine, School of Medicine, University of North Carolina, Chapel Hill, NC 27402, USA; 7Department of Nutrition, Gillings School of Global Public Health, University of North Carolina, Chapel Hill, NC 27402, USA; 8VA Eastern Colorado Geriatric Education and Clinical Center, Rocky Mountain Regional VA Medical Center, Aurora, CO 80045, USA; 9Thurston Arthritis Research Center, University of North Carolina, Chapel Hill, NC 27599, USA; 10Lemieux Library, Seattle University, Seattle, WA 98122, USA; jbodley@seattleu.edu; 11North Carolina Center for Health and Wellness, University of North Carolina, Asheville, NC 28804, USA; 12Division of Rheumatology, Allergy and Immunology, Department of Medicine, School of Medicine, University of North Carolina, Chapel Hill, NC 27599, USA; 13School of Nursing and Health Studies, Monmouth University, West Long Branch, NJ 07764, USA

**Keywords:** nutrition, arthritis, dietary patterns, osteoarthritis symptoms

## Abstract

While there is some research investigating whole foods or diets that are easily understood and accessible to patients with osteoarthritis, specific nutrients or nutraceuticals are more commonly identified. Unfortunately, guidelines and evidence surrounding individual nutrients, extracts, and nutraceuticals are conflicting and are more difficult to interpret and implement for patients with osteoarthritis. The purpose of this umbrella review is to provide a comprehensive understanding of the existing evidence of whole foods and dietary patterns effects on osteoarthritis-related outcomes to inform evidence-based recommendations for healthcare professionals and identify areas where more research is warranted. A literature search identified relevant systematic reviews/meta-analyses using five databases from inception to May 2022. Five systematic reviews/meta-analyses were included in the current umbrella review. Most evidence supported the Mediterranean diet improving osteoarthritis-related outcomes (e.g., pain, stiffness, inflammation, biomarkers of cartilage degeneration). There was little to no evidence supporting the effects of fruits and herbs on osteoarthritis-related outcomes; however, there was some suggestion that specific foods could potentiate symptom improvement through antioxidative mechanisms. The overall lack of homogeneity between the studies limits the conclusions that can be made and highlights the need for quality research that can identify consumer-accessible foods to improve osteoarthritis-related symptoms.

## 1. Introduction

Osteoarthritis (OA) is the most common rheumatic disease, affecting over 32.5 million adults in the United States. Prevalence continues to increase as the population ages and grows; the Centers for Disease Control and Prevention currently predicts that an estimated 75 million US adults will have clinically diagnosed arthritis by 2035. OA is a leading cause of disability, affecting activities of daily living and quality of life [1]. To date, there are no curative or effective disease-modifying medications for the treatment of OA [2]. Interventional strategies often focus on patient education [3] and risk factor modification (e.g., physical therapy, diet, weight loss, regular exercise) to slow progression [4,5], manage symptoms [5], and improve function and quality of life [2,5,6,7,8,9]. There is little distinction between treating OA illness (i.e., pain, symptoms, patient-reported outcomes) and OA disease (i.e., radiographic evidence of OA) [10]; however, treating OA illness is imperative to decrease the years lived with disability and improve the quality of life for individuals with OA. In particular, nonpharmacological strategies and lifestyle modifications (e.g., dietary, physical activity, rehabilitation) are strongly recommended by the American College of Rheumatology to improve OA-related outcomes and reduce disability [2]. Regular exercise can improve OA-related outcomes through weight loss and reduced joint loading [11], decreased systemic inflammatory biomarkers linked to deleterious cartilage effects [12], and sufficient and proper loading of articular cartilage, which is imperative for maintaining tissue health [13]. Early rehabilitation and physical therapy interventions have been shown to improve function and quality of life in patients with OA [8,14]. Research also shows that healthy dietary patterns and diet interventions mitigate OA progression [4], decrease inflammatory markers that hasten cartilage metabolism [15,16], and yield reductions in body mass [9].

As early as Hippocrates, food, particularly a healthy diet, has been viewed as critical to maintaining good physical and emotional health [17]. Poor dietary quality or diets with higher inflammatory potential are linked to accelerated OA progression or incidence of symptomatic OA [4]. For example, the typical Western diet is high in saturated fat and refined carbohydrates, which induces a variety of health consequences via increased adipocyte dysfunction and inflammatory signaling cascades. Although the Western diet is strongly correlated with OA symptoms, evidence is less conclusive surrounding other dietary patterns and OA outcomes [18]. Nutraceuticals (e.g., glucosamine, chondroitin) have been associated with some delay in the progression of knee OA but require pharmaceutical-grade preparation and strict regulatory control for adequate benefit [19]. Specific nutrients and spices (e.g., vitamins D and K [20,21,22], ginger [23,24,25], turmeric [24]) are also associated with improvement of OA symptoms and quality of life. While less studied, diet quality, dietary patterns, and food groups can positively impact OA-related outcomes [4,18].

Individuals with OA believe that diet plays a critical role in their physical health and symptoms [26], and prefer self-management and nutritional education pertaining to foods that should be avoided or targeted based on OA symptom impact [27]. Specifically, patients with OA indicate that they are interested in learning about berries, nuts, herbs, spices, and the Mediterranean diet [27]. While there is some research investigating whole foods or diets that are easily understood and accessible to patients with OA, most research has identified specific nutrients or nutraceuticals that may improve OA symptoms. Unfortunately, guidelines and evidence surrounding individual nutrients, extracts, and nutraceuticals are conflicting and provide granular data that are difficult for patients with OA to interpret and implement. Wholistic dietary pattern assessments and food-consumption frequency is a valid method in epidemiological nutrition research to better understand the effects of dietary patterns rather than assessing the consumption of specific micronutrients, as it is difficult for patients, clinicians, and researchers to quantify day-to-day micronutrient consumption [28,29]. Leveraging food-frequency and dietary pattern assessments to determine associations between dietary patterns and OA-related outcomes may be an approach to overcome the clinical barriers of assessing micronutrient consumption and therefore yield patient- and provider-accessible evidence [4]. However, despite the promising direction of nutritional epidemiology in OA research, and patients reporting interest in nutritional education surrounding whole foods and diet [27], there is limited literature that provides a comprehensive overview of findings related to diet and OA management. Thus, there remain a critical scientific gap and clinical barrier to progressing the goal of reducing the overall burden of OA through underlining the evidence-based dietary patterns that improve OA-related outcomes.

Therefore, the purpose of this umbrella review is to better understand existing evidence and provide a comprehensive review surrounding the effects of whole foods and diets on OA-related outcomes. The results of this umbrella review are intended to inform evidence-based practice, improve OA self-management options, minimize the clinical barrier of interpreting research on extracts and nutraceuticals, and highlight areas where further research is warranted.

## 2. Methods

### 2.1. Protocol and Search Strategy

This review was conducted in accordance with the Preferred Reporting Items for Systematic Reviews and Meta-Analyses (PRISMA) guidelines [30]. Five databases (Medline (PubMed), CINAHL (EBSCO), Web of Science, Cochrane Library (Wiley), and Embase (Ovid)) [31,32] were utilized to search the relevant literature from inception to June 2022. Briefly, the searches included key concepts and search terms relevant to eating plans and nutrition (e.g., Mediterranean diet), whole foods (e.g., berries, vegetables, nuts), OA, and OA-related outcomes (e.g., patient-reported outcomes, symptoms, inflammation). A reference librarian (JB) assisted with the design and protocol of the full search strategy across the 5 databases. The full search strategy and search terms can be found in the Appendix A.

### 2.2. Eligibility Criteria and Selection of Studies

The full eligibility criteria can be found in Table 1. Briefly, article titles, abstracts, and full texts were screened based on population (adults with hand, hip, or knee OA), intervention (dietary and/or whole-food intervention), comparison (no diet or normal practice), outcomes (pain and/or disease incidence/progression), study design (systematic review or meta-analysis). All authors contributed to the development of the eligibility criteria.

Two authors (ANB and JB) conducted the literature search across the 5 databases in May 2022, and all articles (*n* = 1245) were imported into EndNote 20.3 (Berkely, CA, USA). Duplicates were then removed (*n* = 439). One author (ANB) screened the remaining articles (*n* = 806) and removed nonrelevant articles based on title (*n* = 635). Next, the same author (ANB) screened the abstracts of the remaining articles (*n* = 171) and removed nonrelevant articles (*n* = 142). Lastly, a full-text screen of the remaining articles (*n* = 29) was completed to identify relevant articles to be included in the present review (*n* = 5; Figure 1). Common reasons for exclusion after title, abstract, and full-text screens included: the systematic review or meta-analysis only assessed extracts of whole foods or spices/herbs in pill form; review comparison groups did not meet inclusion criteria (e.g., exercise and diet, nutrition education, or exercise alone); exclusively assessed inflammatory arthritis (e.g., rheumatoid, psoriatic); or the review did not provide sufficient detail in methodology to ensure that the inclusion criteria of the current umbrella review were met. 

### 2.3. Data Extraction

Data extraction was completed independently by 2 authors (JAB and CBN). A third author (SPS) subsequently reviewed the extracted data and identified any duplicate original research studies between the systematic reviews. The data were compiled in a Microsoft Excel Spreadsheet (Version 16.68; Microsoft, Redmond, WA, USA). All authors discussed trends within the extracted data in order to interpret the findings.

### 2.4. Critical Appraisal

Critical appraisals of each article were completed independently by 3 researchers (LMA, LFC, HKV). The Joanna Briggs Institute Critical Appraisal Checklist for Systematic Reviews and Research Syntheses [33] framework was utilized for the critical appraisal (11 questions scores as yes = 1, no/unclear = 0). Each review received a score of either 1 or 0 on each of the 11 questions, for a total of 11 possible points. A score of 1 (yes) was given if the review sufficiently met the requirements outlined in the question; if the review did not explicitly meet the requirements outlined in the question, or it was unclear whether the requirements were met, a score of 0 (no/unclear) was given for that question. If the criteria outlined in the question were not applicable to the review, a score of “NA” was given, and that question was not considered in the final scoring of the review article (i.e., the review was scored out of 10 points rather than 11 if it received an “NA” for one question). Critical appraisal scores are reported as the percentage of “yes” scores out of the total possible for the review based on the scoring criteria described above (Figure 2). After the 3 researchers completed the critical appraisal for each review, a fourth author (ANB) then compiled all critical appraisal scores and resolved any conflicts between the scores.

## 3. Results

Our literature search yielded 1245 articles, of which five met the inclusion criteria of this umbrella review [34,35,36,37,38]. The reasons for exclusion and the final determination of the reviews included are provided in Figure 1.

### 3.1. Methodological Quality Appraisal Results

Critical appraisals of the systematic review articles are provided in Figure 2. The overall quality of the articles ranged from high [35,37] to low [38]. Only one review had no critical weaknesses [35], while three reviews had two or more critical weaknesses [34,36,38]. Four studies used appropriate criteria for appraising studies and assessed likelihood of publication bias [34,35,36,37]. Two studies leveraged adequate sources and resources in the searches [35,36]. Finally, two studies used two or more reviewers independently to appraise the literature [35,37], made efforts to minimize bias or systematic errors during the systematic review [35,37], and utilized appropriate methods to combine studies [35,37].

### 3.2. Review Characteristics

The five review articles included in the present umbrella review focused on relationships between OA-related outcomes (e.g., symptoms, inflammation, management), low inflammation dietary patterns, anti-inflammatory foods (e.g., powders, purees, juices, and tea) [34,35,36,38], and food supplements (e.g., herbs) [37]. An overall summary of the systematic reviews is provided in Figure 2. A total of nine cross-sectional studies and 69 randomized control trials (RCTs)/prospective designs were included in these five reviews. Data for cross-sectional studies were derived from datasets and cohorts such as the Osteoarthritis Initiative (OAI) [34,38], Framingham Osteoarthritis Cohort Study [36], Melbourne Collaborative Cohort Study [36], and the Korean National Health and Nutrition Examination Survey [36].

The five included articles systematically reviewed from three to 52 original research articles, with sample sizes ranging from *n* = 468 to *n* = 185,329. Eighty percent (4/5) of the articles followed the Preferred Reporting Items for the Systematic Review and Meta-Analyses (PRISMA) guidelines, and 60% (3/5) were registered with the International Prospective Register of Systematic Review (PROSPERO). Reviews included either OA and other rheumatoid/osteoarticular conditions combined [35,38] or OA alone [34,36,37]. Original articles focused predominantly on knee OA, with few that focused on hip, hand, or “unspecified” OA. Most of the summarized articles that reported sex distribution studied both sexes and ages ranged from 25 to 90 years.

Systematic review methodology varied widely between the five included articles. Two reviews reported overall numbers of participants allocated into treatment and control groups [35,37], whereas the remaining three did not [34,36,38]. These same two reviews adequately described the article pooling methods that were performed prior to quantitative analysis; pooling was done either by study type [35] or by time point with interventions [37]. Only two of the five reviews conducted a methodological assessment using either GRADE criteria or the Newcastle–Ottawa Scale (NOS) [34,35]. Two reviews provided standardized mean differences or effect sizes [35,37] for nutritional interventions, whereas three provided relative change or general patterns of change in selected outcomes [34,36,38].

### 3.3. Overview of Main Findings

Table 2 provides an overview of the extracted data relevant to the umbrella research question. Findings were grouped by diets and nutrition (Mediterranean diet; Table 3(A)) or food (i.e., fruit, Table 3(B); herbs, Table 3(C)). Figure 3 provides a visual overview of the findings from the data extracted from eligible studies.

Mediterranean diets were explored using 12–24-week prospective or interventional study designs and study cohorts and assessed a variety of OA-related outcomes (i.e., self-rated pain and functional outcomes, objective joint range of motion, OA progression, inflammatory biomarkers). It should be noted that several papers on Mediterranean diet were derived from a single study (COHORT) [39,40,41,42,43] and summarized in more than one systematic review. Very small changes were observed with self-rated function using Medical Outcomes Short Form (SF)-12, Western Ontario McMaster University Index (WOMAC), and Arthritis Impact Measurement Scale (AIMs) scores [40,44]. Objective measures of knee and hip mobility improved [44]. Mediterranean diet interventions were associated with modest reductions in cartilage degradation [44], reductions in inflammatory biomarkers (i.e., interleukin-6 [IL-6]) [44], and reductions in overall OA pain derived from various assessments [44]. Cohort data from the OAI revealed associations between adherence to Mediterranean diet and improvements in SF-12 Physical Component scores [40], lower WOMAC disability score, lower prevalence of knee OA [39,40], and lower WOMAC pain severity [40]. Mediterranean diet adherence score determination included an unweighted classification of the frequency of consumption of ‘beneficial’ food group contributors to this diet pattern based on recall (e.g., vegetables, legumes, fruits, and cereals) relative to the consumption of ‘detrimental’ contributors (e.g., dairy, meat). As such, this measure provides generalized, large population associations but does not provide mechanistic-specific insight regarding the diet’s direct effect on OA.

Evidence relating to fruit intake and OA outcomes was derived from studies assessing tart cherry juice [45], pomegranate juice [46], and strawberry powder [47]. Study designs included crossover and parallel prospective designs with controls or placebos. With respect to physical function and mobility, only strawberry powder appeared to reduce self-reported Health Assessment Questionnaire scores [47]. Objective measures of mobility and function were not affected by any of the fruit interventions. Longitudinal observational data from a single study indicated that 12 weeks of freeze-dried strawberry powder consumption daily (50 g, which is equivalent to 500 g fresh strawberries) was associated with reduced prevalence of constant OA pain symptoms [47]. Inflammatory biomarkers were differentially affected by fruit intervention type. Specifically, pomegranate juice lowered metalloproteinase matrix protein (MMP)-13 but not MMP-1 [46], and strawberry powder lowered MMP-3 concentrations but not MMP-8 [47]. Tart cherry juice appeared to reduce high-sensitivity C-reactive protein (hs-CRP) levels upon initial introduction to the intervention [45], but strawberry powder had no effect on hs-CRP [47]. Finally, strawberry powder reduced interleukin-6 and interleukin-1β levels over 12 weeks [47]. Collectively, these findings suggest that these fruit interventions may induce some anti-inflammatory responses in individuals with OA, but further investigation is warranted given the inconclusive nature of the data. 

**Table 3 nutrients-15-03050-t003:** Summary of extracted data relevant to dietary patterns and foods on OA-related outcomes (function, mobility, inflammation, pain, and OA progression) for Mediterranean diets (A), fruit (B), and herbs (C).

**(A). Mediterranean Diet**				
**Author, Year**	**Systematic Review**	**Population**	**Intervention and Comparison**	**Outcome: Function and Mobility**
Dyer, 2017 [44]	Genel [35] Sanpaolo [38]	*n* = 99 Nonspecific OA	Mediterranean diet and regular diet RCT 16 weeks	Change in AIMS physical component scores: −0.1 ± 0.29 pts Mediterranean diet vs. −0.1 ± 0.38 Regular diet (*p* < 0.05); standard mean difference 0.0 (95% CI −0.40, 0.40); increased knee flexion and hip rotation ROM in Mediterranean diet (*p* < 0.05).
Veronese, 2017 [40]	Morales-Ivorra [34]	*n* = 4470 OAI cohort	Adherence to Mediterranean diet score No comparison group	Adherence scores were positively related to SF-12 Physical component scores (*p* = 0.0001); low adherence to vegetable intake increased odds of low SF-12 PCS score (OR 1.52; *p* = 0.01); Adherence was also related to lower WOMAC disability scores (right knee −0.08 (95% CI −0.14, −0.03) and left knee −0.07 (95% CI −0.1, −0.01); *p* < 0.05).
				**Outcome: OA Progression**
Dyer, 2017 [44]	Genel [35]	*n* = 99 Nonspecific OA	Mediterranean diet and regular diet	Markers of cartilage degradation decreased by 8% with Mediterranean diet (*p* = 0.014)
Veronese, 2016 [39]	Morales-Ivorra [34]	*n* = 4470 OAI cohort	Adherence to Mediterranean diet score No comparison group	Adherence to a Mediterranean diet decreased OA prevalence; highest adherence score had lower prevalence of knee OA vs. lowest adherence quartile (25.2% vs. 33.8% prevalence; *p* < 0.0001)
Veronese, 2017 [40]	Sanpaolo [38]	*n* = 4358 OAI cohort	Adherence to Mediterranean diet score No comparison group	OR for knee OA was lowest in highest adherence score quartile (OR = 0.83 (95% CI: 0.69, 0.99); *p* < 0.04); increased adherence to Mediterranean diet was associated with decreased OA prevalence
				**Outcome: Pain and Stiffness**
Dyer, 2017 [44]	Genel [35]	*n* = 99 Nonspecific OA	Mediterranean diet and regular diet RCT 16 weeks	Change in pain score: −0.3 points Mediterranean diet vs. −0.6 points regular diet (SMD = 0.56 (95% CI: 0.15, 0.96))
Veronese, 2017 [40]	Morales-Ivorra [34]	*n* = 4470 OAI cohort	Adherence to Mediterranean diet score No comparison group	Adherence to a Mediterranean diet associated with lower WOMAC pain subscale scores (right knee: −0.02 (95% CI: −0.04, −0.01); left knee: −0.02 (95% CI: −0.04, −0.003); *p* < 0.05)
				**Outcome: Inflammation**
Dyer, 2017 [44]	Genel [35]	*n* = 99	Mediterranean diet and regular diet RCT 16 weeks	Change in cartilage degradation biomarker IL-6: −1.76 ± 1.1 units Mediterranean diet vs. −0.22 ± 0.41 units regular diet (*p* < 0.05; SMD= −1.71 (95% CI: −2.34, −1.08))
Dyer, 2017 [44]	Morales-Ivorra [34]	*n* = 4358	Mediterranean diet and regular diet	Most biomarkers in Mediterranean diet decreased by 47% (*p* = 0.01)
**(B). FRUIT**				
**Author, Year**	**Systematic Review**	**Population**	**Intervention**	**Outcome: Function and Mobility**
Goochani, 2016 [46]	Guan [36]	*n* = 38 Knee OA	Pomegranate juice (200 mg) and control Parallel 6 weeks	No effect on WOMAC Function scores
Schell, 2017 [47]	Genel [35]	Knee OA	Freeze-dried strawberry powder (50 g = 500 g fresh strawberries) and placebo powder Crossover 12 weeks	Change in health assessment questionnaire: −0.2 ± 0.4 points with strawberry powder vs. 0.0 ± 0.14 with placebo powder
Schumacher, 2013 [45]	Genel [35]	*n* = 53 Knee OA	Tart cherry juice (>90% with apple juice = 100 cherries) Crossover	No effect on WOMAC Function scores No effect on 10 m walk times
				**Outcome: Pain and Stiffness**
Goochani, 2016 [46]	Guan [36]	*n* = 38 Knee OA	Pomegranate juice (200 mg) and control Parallel 6 weeks	No effect on WOMAC Pain or Stiffness scores
Schell, 2017 [47]	Genel [35]	Knee OA	Freeze-dried strawberry powder (50 g = 500 g fresh strawberries) and placebo powder Crossover 12 weeks	Change in pain scores: −0.6 ± 0.22 points with strawberry powder vs. −0.4 ± 0.28 points with placebo powder (SMD = −0.77 (95% CI: −1.47, −0.07))
Schell, 2017 [47]	Guan [36]	18 yo OA self-reported chronic pain	Freeze-dried strawberry powder (50 g = 500 g fresh strawberries) and placebo powder Crossover 12 weeks	ICOAP experiencing constant pain 13.8% (intervention) and 24.2% (crossover; *p* = 0.02)
Schumacher, 2013 [45]	Guan [36]	*n* = 53 Knee OA	Tart cherry juice (>90% with apple juice = 100 cherries) Crossover	No effect on WOMAC Pain or Stiffness scores
				**Outcome: Inflammation**
Goochani, 2016 [46]	Guan [36]	*n* = 38 Knee OA	Pomegranate juice (200 mg) and control Parallel 6 weeks	Intervention vs. control effects: MMP-13 (127.99 vs. 159.87 pg/mL; *p* = 0.02); glutathione peroxidase (67.0 vs. 81.2 μg/mL^2^; *p* = 0.0001); no effect on MMP-1
Schell, 2017 [47]	Guan [36]	18 yo OA self-reported chronic pain	Freeze-dried strawberry powder (50 g = 500 g fresh strawberries) and placebo powder Crossover 12 weeks	Intervention vs. crossover effects: IL-6 (3.4 vs. 8.7 pg/mL; *p* = 0.0006); IL-1β (7.5 vs. 16.2 pg/mL; *p* < 0.0001); MMP-3 (5.3 vs. 6.8 ng/mL; *p* = 0.004); no effect on hs-CRP and MMP-8
Schumacher, 2013 [45]	Guan [36]	*n* = 53 Knee OA	Tart cherry juice (>90% with apple juice = 100 cherries) Crossover	Intervention effect on hs-CRP: after first treatment (1.98 vs. 4.21 mg/L); after second treatment (3.49 vs. 3.17 mg/L); *p* values not reported
**(C). HERBS**				
**Author, Year**	**Systematic Review**	**Population**	**Intervention and Comparison**	**Outcome: Function and Mobility**
Connely, 2014 [48]	Guan [36]	*n* = 46 Knee OA	High rosA spearmint tea with rosmarinic acid (260–300 mg) Parallel 16 weeks	No significant effect on total WOMAC scores or subscale scores; no significant effect on stair climb test performance; no significant effect on Medical Outcome Short Form 36 general health surveys
				**Outcome: Pain and Stiffness**
Altman, 2001 [23]	Mathieu [37]	*n* = 247 Knee OA	Ginger extract and placebo RCT 6 weeks	SMD in VAS Pain scores: −3.76 points (95% CI: −6.88, −0.65 points) reported
Wigler, 2003 [49]	Mathieu [37]	*n* = 29 Knee OA	Ginger extract (250 mg/day) vs. placebo RCT 12 weeks	SMD in VAS Pain scores: −3.76 points (95% CI: −6.88, −0.65 points) reported
Bolognesi, 2016 [25]	Mathieu [37]	*n* = 54 Knee OA	Ginger vs. placebo RCT 24 weeks	SMD in VAS Pain scores: −3.76 points (95% CI: −6.88, −0.65 points) reported
				**Outcome: Inflammation**
Altman, 2001 [23]	Mathieu [37]	*n* = 247 Knee OA	Ginger extract and placebo RCT 6 weeks	SMD CRP: −1.36 units (95% CI: −1.80, −0.92); mean difference erythrocyte sedimentation rate: −2.13 (95% CI: −3.37, −0.89)
Wigler, 2003 [49]	Mathieu [37]	*n* = 29 Knee OA	Ginger extract (250 mg/day) vs. placebo RCT 12 weeks	SMD CRP: −1.36 units (95% CI: −1.80, −0.92); mean difference erythrocyte sedimentation rate: −2.13 (95% CI: −3.37, −0.89)
Bolognesi, 2016 [25]	Mathieu [37]	*n* = 54 Knee OA	Ginger vs. placebo RCT 24 weeks	SMD CRP: −1.36 units (95% CI: −1.80, −0.92); mean difference erythrocyte sedimentation rate: −2.13 (95% CI: −3.37, −0.89)

AIMS = Arthritis Impact Measurement Scale; ROM = range of motion; OAI = Osteoarthritis Initiative; OR = odds risk; CI = confidence interval; DAS28 = disease activity index; WOMAC = Western Ontario McMaster University Index; ICOAP = intermittent and constant osteoarthritis pain; MMP = matrix metalloproteinase; IL = interleukin; hs-CRP = high-sensitivity C reactive protein; VAS = visual analogue scale; SMD = standardized mean difference.

Evidence of herb consumption on OA outcomes is limited. Data from four eligible studies assessed spearmint tea or ginger interventions for knee OA [23,25,48,49]. Spearmint tea intake over 16 weeks did not produce improvements in subjective measures (WOMAC scores, SF-36 scores) or objective physical function and mobility (walk test, stair climb performance) [48]. Findings from three RCTs [23,25,49] ranging from 6 to 24 weeks comparing ginger versus placebo were pooled to provide standardized mean differences in visual analogue scale pain scores, CRP levels, and erythrocyte sedimentation rate; however, the individual study results were not provided. Although firm conclusions are not possible, these summary scores suggest that ginger may decrease knee OA pain severity and specific inflammatory biomarker levels [23,25,49].

## 4. Discussion

The findings across the reviews included in the present umbrella review suggest that the Mediterranean diet, fruit, and herbs may improve OA-related outcomes. Importantly, most data suggested that these interventions did not worsen OA-related outcomes or OA progression. The heterogeneity and overall lack of methodologically strong studies limit conclusions that can be made. As such, the current umbrella review highlights the need for further research evaluating the effects of whole foods and diets on both OA illness (i.e., patient-reported outcomes, symptoms, function), disease (i.e., radiographic evidence of OA), and OA progression.

The Mediterranean diet was the most commonly studied eating plan across the reviews. Previous findings suggest that dietary patterns consistent with the Mediterranean diet are associated with better OA management [4], and patients with OA are most interested in learning about the Mediterranean diet and influences on OA symptoms [27]. Evidence suggests that following a Mediterranean-style food plan improves both OA illness and disease as measured by patient-reported outcomes, pain, symptoms, mobility, cartilage degeneration, and inflammatory biomarkers [34,38]. Although definitions of Mediterranean-style food plans vary, most plans consist of high quantities of fruits, vegetables, legumes, nuts, and seafood, and moderate consumption of dairy, olive oil, and poultry [50]. Such dietary patterns have the capability to reduce systemic inflammation and may influence the incidence and progression of OA. Overweight and obesity increases risk for OA via chronic, low-grade inflammation as well as biomechanical joint overloading [51]. The Mediterranean diet reduces systemic inflammation [52] and yields favorable weight loss results in individuals with overweight and obesity [53,54]. Therefore, while the favorable findings for the impact of the Mediterranean diet on OA in the present umbrella review were partially expected, these data also highlight the multifactorial nature of OA and how two major components of disease may be modified via dietary interventions.

The included systematic reviews suggest that specific foods such as strawberries and ginger could potentiate symptom improvement through antioxidative mechanisms. Antioxidants work to decrease the presence of reactive oxygen species, which have a damaging effect on the surrounding tissues over time and contribute to inflammation [55]. Strawberries, for example, are a fruit that is high in Vitamin C (ascorbic acid) with a range of 0.04–0.07 g per 100 g [56]. Vitamin C is an essential cofactor for α-ketoglutarate-dependent dioxygenases, such as prolyl hydroxylases, which contributes to the biosynthesis of collagen in addition to being a water-soluble antioxidant [57,58]. Ginger, a common flavorful ingredient in cooking, contains compounds such as gingerol and shogaol that reduce the expression of cytokines tied to inflammation such as IL-1, IL-6, and TNF-alpha [59]. While strawberries and ginger contain the antioxidants needed to reduce inflammation and subsequently improve pain severity associated with OA, the literature does not provide conclusive evidence of the efficacy of these specific whole foods in the prevention and management of OA.

Previous research has largely focused on how specific dietary supplements may influence OA disease and illness [20,21,22,60], yet these data are difficult for both patients and healthcare providers to quantify in daily dietary patterns. Further, the granularity of supplement and nutraceutical data and findings require a certain level of scientific and health literacy to successfully interpret and implement, which is a critical clinical barrier. The purpose of this umbrella review was to provide a comprehensive overview of accessible nutritional patterns that patients are interested in learning about [27] to inform evidence-based practice for improving OA-related outcomes through behavior modification such as diet and nutrition. Even though whole-food selections and dietary modifications might be understandable and accessible to patients, data and systematic reviews which assess the influence of these modifications on OA disease and illness are very limited. One potential factor contributing to the limited data surrounding whole foods and diet on OA may be the difficulty in designing and conducting RCTs that can accurately quantify the studied nutrients within whole foods, standardize the amount of the nutrient, and effectively blind the participant to the supplement or placebo. Moreover, there are challenges in interpreting clinical findings and impact based on the numerous and unique interactions of nutrients from whole foods combined in a diet together. This tension between research versus ecological validity severely constrains conclusive research that is needed to inform healthcare providers in their evidence-based decision making. However, a reductionistic nutritional epidemiology approach that leverages validated food-frequency assessments [17,28,29] may be useful in identifying whole food and dietary patterns that are linked to better OA outcomes and subsequently elucidate potential micronutrient interactions to be further investigated in RCTs. The current umbrella review sought to provide a broad and comprehensive assessment of current review literature surrounding the associations between dietary patterns and OA-related outcomes, and serve as an initial step in providing evidence-based findings for both patients and healthcare providers. 

## 5. Limitations

While the present umbrella review is one of the first to assess and summarize the findings of systematic and meta-analysis reviews on the effects of diet and whole foods on OA-related outcomes, a few limitations exist. First, in contrast to granular data abstracted for a systematic review or meta-analysis, umbrella reviews are meant to quickly ascertain large amounts of data to allow a comparison of high-level findings. Such reviews can be conducted quickly and are critical for the establishment of new policies or practices. The ability to evaluate a large body of information provides a comprehensive summary highlighting the current evidence base [61]. However, as individual study-level data are not abstracted, there is a possibility that lower methodological quality could not be identified by this umbrella review. Second, combining reviews with different criteria increases the risk of heterogeneity, as interventions and inclusion/exclusion criteria may differ. For example, most of the included studies evaluated joint pain, but few evaluated OA progression, joint function, or mobility. In some studies, the OA-related outcome was not specified. Interpretation may also be limited due to small sample sizes, inclusion of various arthritis types, and a lack of uniformity among measured outcomes (i.e., biomarkers, functional status, and measures of pain, quality of life, and physical activity).

## 6. Conclusions

Overall, the limitations of studies investigating effects of nutrition and whole food, particularly those that include fruit and herbs, on osteoarthritis indicate a strong need for more research in larger study populations using an RCT design. Studies of consumer-accessible foods (i.e., whole foods and diets), rather than powders, extracts, or supplements are needed to inform evidence-based practice and provide realistic and attainable recommendations for dietary patterns that may improve OA outcomes. This research is difficult to complete due to the tension between research and ecological validity that is associated with measuring whole foods. However, it is critical to the evidence-based practice of healthcare providers that this research be prioritized.

## Figures and Tables

**Figure 1 nutrients-15-03050-f001:**
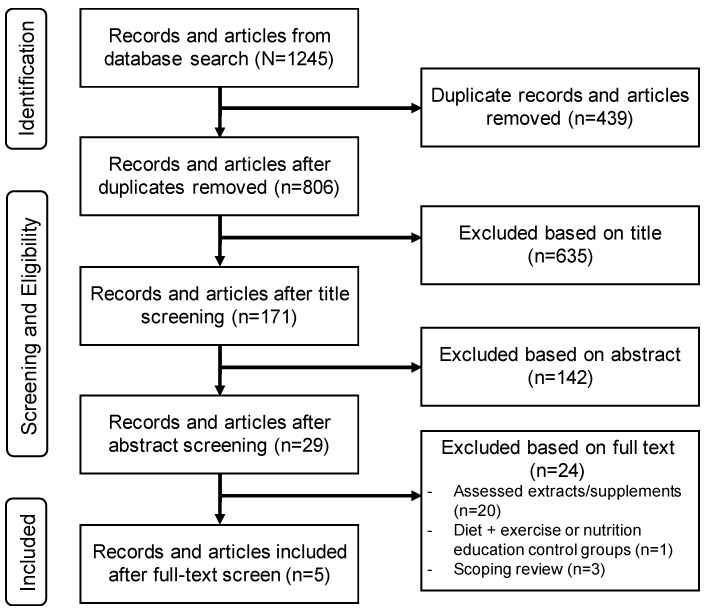
Preferred Reporting Items for Systematic Reviews and Meta-Analyses (PRISMA) flowchart of articles screened for eligibility.

**Figure 2 nutrients-15-03050-f002:**
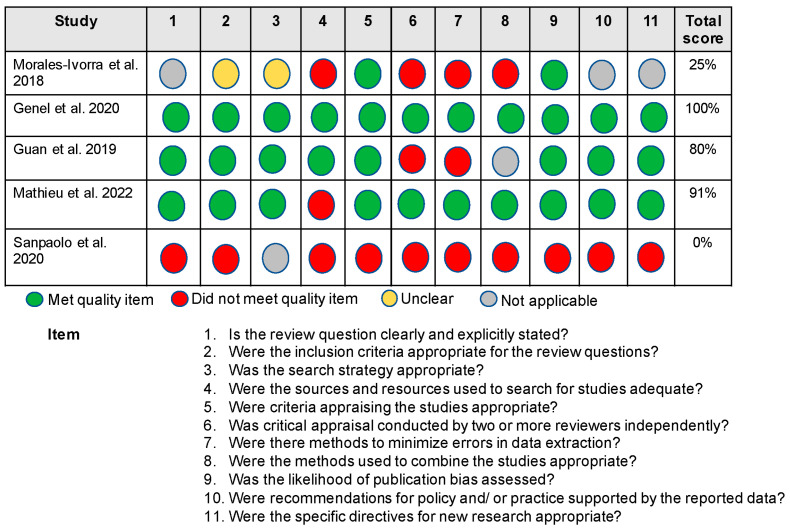
Critical Appraisal of the Evidence [34,35,36,37,38].

**Figure 3 nutrients-15-03050-f003:**
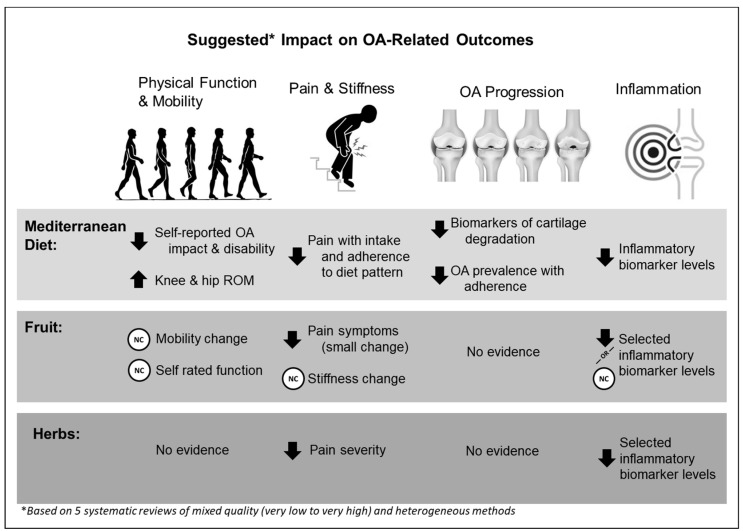
Overview of the findings from the data extracted from eligible studies on OA outcomes of physical function and mobility, disease progression, pain and stiffness, and inflammation. Arrows indicate direction of change; NC indicates no change. ROM = range of motion; OA = osteoarthritis.

**Table 1 nutrients-15-03050-t001:** Eligibility criteria of included studies.

**Population**	18+ years old; physician-diagnosed and/or radiographic evidence of hand, hip, or knee OA
**Intervention**	Association between diet/whole foods and OA symptoms/outcomes
**Comparison**	Compared to no diet and/or usual diet and/or NSAIDS and/or therapy and/or usual practice (medication, exercise, physical therapy)
**Outcomes**	*Primary outcomes:* OA-related pain/stiffness and progression/incidence (incident radiographic, joint space narrowing, cartilage loss/damage) *Secondary outcomes:* inflammation, physical function, mobility, joint stiffness
**Study Design**	Systematic reviews and meta-analyses

OA = osteoarthritis; NSAIDs = nonsteroidal anti-inflammatory drugs.

**Table 2 nutrients-15-03050-t002:** Summary Characteristics of Included Reviews.

Review, Date Range	N Studies, Participants, Demographics	Countries	Study Design	Methods, Quality Analysis, and Review Goal
**Genel et al., 2020 [35]** Up to July 2019	7 studies 468 participants 42.8% knee OA or unspecified OA 71.4% included males and females 28.6% included females only 30–90-yr age range	Germany USA England, UK Mexico Sweden	2 prospective pre-post 5 RCT design	Followed PRISMA; PROSPERO-registered, GRADE assessment of evidence; 7 studies in qualitative synthesis; 5 studies in quantitative, meta-analysis; bias risk via Cochrane Handbook’s ROB Version Checklist and ROBINS-I Tool; forest plots and I^2^ index for between-study heterogeneity reported; PICOS framework used to select studies and intention-to-treat data were preferentially analyzed Goal: to determine health impact of low inflammatory diet (whole foods or supplements) on inflammatory biomarkers, joint symptoms, quality of life, and weight change
**Guan et al., 2019 [36]** January 2004–March 2016	9 studies 6915 participants in cross-sectional 75% knee OA 25% hip OA 100% included males and females 176 participants in RCT 100% knee OA 100% included males and females	USA Australia Korea Iran Japan	4 cross-sectional 3 parallel design 2 crossover design	Followed PRISMA; PROSPERO-registered; bias risk via Cochrane Handbook’s ROB Version 2 checklist and ROBINS-I Tool Goal: to determine the effects of dietary phytochemical intake from foods on progression of OA in adult humans
**Mathieu et al., 2022 [37]** Up to November 2021	52 studies 4744 participants in RCT 3.9% general OA 1.9% hip OA 94.2% knee OA 71.1–73.2% were females in treatment and control groups	Not listed	52 RCT	Followed PRISMA; PROSPERO-registered; bias risk via Jadad scale; forest plots, confidence intervals, I^2^ index for between-study heterogeneity reported; robustness of results tested with sensitivity analyses; subgroup analyses performed for different nutritional interventions Goal: to assess the effects of nutrients and vitamins on symptoms in patients suffering from OA
**Morales-Ivorra et al., 2018 [34]** Up to December 2020	3 studies 5131 participants in cross-sectional OAI database Mixed OA sites or knee OA only 58.1% females 124 participants in RCT Knee, hip, finger OA 83% females 30–90-yr age range	USA England, UK	2 cross-sectional 1 RCT	Followed PRISMA; bias risk via Newcastle–Ottawa Scale (NOS) for quality on nonrandomized studies in meta-analyses; comparison of effects of diet on prevalence of OA and OA-related symptoms, quality of life, cartilage degeneration, and inflammation Goal: to review and analyze epidemiological studies to find the associations between Mediterranean diet and OA
**Sanpaolo et al., 2020 [38]** Up to January 2018	70 studies 9028 participants in cross-sectional OAI database Mixed OA or knee OA only 76,301 participants in RCT or prospective studies 25–90-yr age range	USA Spain France Sweden Scotland	3 cross-sectional 2 RCT 2 prospective cohort	A summary of the studies included in the review was provided with a description of associations between diet and prevalence of OA and OA-related symptoms and quality of life Goal: to determine the beneficial effects of Mediterranean diets on osteoarticular disease

RCT = randomized controlled trial; PRISMA = Preferred Reporting Items for the Systematic Review and Meta-Analyses; PROSPERO = International Prospective Register of Systematic Review; PICOS = patient/population, intervention, comparison, and outcomes; OAI = Osteoarthritis Initiative; OA = osteoarthritis.

## Data Availability

No new data were created or analyzed in this study. Data sharing is not applicable to this article.

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
