# Peer review of "Evidence-Based Dietary Practices to Improve Osteoarthritis Symptoms: An Umbrella Review"

_nutrients, 2023, doi:10.3390/nu15133050_

Round 1

Reviewer 1 Report

The present systematic review assesses the evidence-based dietary practices to improve osteoarthritis symptoms. The topic is relevant, but certain deficiencies identified in both content and form need to be addressed based on the specific recommendations below:

Keywords must reflect the main characteristic words of the paper (usually reflected also by the title) in the best way to increase the paper's relevance and chances of being found when searching for it after keywords. Therefore, I suggest improving this section.

L48-49; L58-L62 there should be only one bibliographic index because it is continuous information.

L49-54 there is no bibliographic index. Bibliographic references should be added.

L66- too many bibliographic resources for such short information It should be detailed, and bibliographic references should be added after each specific nutrient or spice.

The aim of the paper should be separately approached in the last paragraph of the introduction section, along with the contributions to the field and the novelty of the paper.

The introduction is poorly addressed in the current paper and needs to be improved with updated and scientifically validated information. The intervention strategies mentioned in L51 need to be detailed for a clearer overview, and recovery management needs to be added as it becomes essential for the overall management of the pathology and increasing the quality of life of patients as the disease is not curable. I suggest checking and referring to: PMID: 35454333.

The result sections should not have any bibliographic indexes. Any comment, evaluation, or comparison with other research is made in the discussion section.

Table 1 - a table head must be designed and inserted.

What does it mean to be "excluded based on title" in the PRISMA chart? It should be clarified.

The selection of those databases should be explained since there are many databases with different advantages.

Reference number 1- it should include the date of access.

Reviewer 2 Report

The study is a comprehensive umbrella review that synthesizes existing literature to examine the effects of whole foods and diets on osteoarthritis (OA) outcomes. The main findings suggest that following a Mediterranean diet, consuming fruits, and using herbs might improve OA-related outcomes, without worsening the disease or its progression. The Mediterranean diet, in particular, shows promising effects, likely due to its anti-inflammatory properties and its influence on weight management, two key factors in managing OA.

The article is well articulated with several strong points. The study conducts a comprehensive analysis of existing literature, allowing it to capture a wide range of findings and present a more complete understanding of the relationship between diet, whole foods, and osteoarthritis outcomes. The authors used multiple databases for their literature search, capturing all relevant literature on the topic. By highlighting the heterogeneity and methodological weaknesses in the existing literature, the review identifies areas where further research is needed, potentially guiding future studies.

One suggestion would be on details of micro neutrients; A key challenge in studies of whole food consumption is accurately quantifying nutrient intake and standardizing the amount of nutrient consumed. The authors acknowledge this challenge but don't provide a solution, which leaves a significant gap for future research. It would be good to provide an opinior in this regard. 

Reviewer 3 Report

Dear Auhors,

thank you for this interesting umbrella review on the exciting topic of osteoarthritis ad diet interventions. An umbrella review aims to summarize and evaluate the findings of multiple systematic reviews and meta-analyses on a specific topic, trying to inform evidence based recomendations for healthcase professionals.

This scientific review provides a comprehensive analysis of the effects of diet and whole foods on osteoarthritis (OA)-related outcomes based on an umbrella review of systematic and meta-analysis reviews. The review focuses on the Mediterranean diet, fruit interventions, and herb consumption, highlighting their potential benefits in improving OA symptoms and disease progression. It also discusses the limitations of current research and the need for further investigation.

The review considerates the quality and relevance of the included systematic reviews and describes them very well. However, there are some missing points, which I would like to point out, and the authros should further explain them:

1. I find the number of the included studies - systematic rewiews to be too low - 5. On my quick search in pubmed with keywords "osteoarthritis diet" there are 254 rewies and systematic rewievs found. Any reasons why your review included, if I have understood correctly, only 5 rewiews? Of which reason were 24 studies in the last step excluded based on the full text? Please explain.

2. There are many modern diets, including paleo, keto diet, frutarians diet.... Please explain in the review, why did you only include Mediterranean diet? (are there no/insuffucient studies for other diets?)

Thank you.

Round 2

Reviewer 1 Report

The authors have significantly improved the manuscript based on the suggestions received.

Reviewer 3 Report

Dear authors,

I kindly suggest that an umbrella review should encompass a wider range of diets, not solely focusing on the Mediterranean diet. It would greatly enhance the value of the review if it includes an examination of other dietary patterns as well. At least the ketogenic and calorie restriction diet with fasting should be included. By incorporating a diverse set of diets, the review will provide a more comprehensive and robust analysis of the topic.

Thank you for considering this suggestion.

Best regards,